# Dimethyl Dicarbonate as a Food Additive Effectively Inhibits *Geotrichum citri-aurantii* of Citrus

**DOI:** 10.3390/foods11152328

**Published:** 2022-08-04

**Authors:** Shuqi Liu, Deyao Zhang, Yuqing Wang, Fan Yang, Juan Zhao, Yujie Du, Zhonghuan Tian, Chaoan Long

**Affiliations:** Key Laboratory of Horticultural Plant Biology of Ministry of Education, Huazhong Agricultural University, National R&D Center for Citrus Preservation, National Centre of Citrus Breeding, Wuhan 430070, China

**Keywords:** postharvest disease, reactive oxygen species, mechanisms, membrane integrity, defense-related enzymes

## Abstract

Dimethyl dicarbonate (DMDC), a food additive, can be added to a variety of foods as a preservative. This study aimed to evaluate the inhibitory effects of DMDC on *Geotrichum citri-aurantii* in vitro and in vivo, as well as the potential antifungal mechanism. In vitro experiments showed that 250 mg/L DMDC completely inhibited the growth of *G. citri-aurantii* and significantly inhibited spore germination by 96.33%. The relative conductivity and propidium iodide (PI) staining results showed that DMDC at 250 mg/L increased membrane permeability and damaged membrane integrity. Malondialdehyde (MDA) content and 2, 7-Dichlorodihydrofluorescein diacetate (DCHF-DA) staining determination indicated that DMDC resulted in intracellular reactive oxygen species (ROS) accumulation and lipid peroxidation. Scanning electron microscopy (SEM) analysis found that the mycelia were distorted and the surface collapsed after DMDC treatment. Morphological changes in mitochondria and the appearance of cavities were observed by transmission electron microscopy (TEM). In vivo, 500 mg/L DMDC and *G. citri-aurantii* were inoculated into the wounds of citrus. After 7 days of inoculation, DMDC significantly reduced the disease incidence and disease diameter of sour rot. The storage experiment showed that DMDC treatment did not affect the appearance and quality of fruits. In addition, we found that DMDC at 500 mg/L significantly increased the activity of citrus defense-related enzymes, including peroxidase (POD) and phenylalanine ammonia-lyase (PAL). Therefore, DMDC could be used as an effective method to control citrus sour rot.

## 1. Introduction

Citrus is the most productive fruit in the world. The global production of citrus fruit exceeded 158 million tons in 2020. Brazil, China, and the United States are the leading producers (FAO, 2020). Citrus fruit has become popular worldwide because of its delicious taste and nutrient richness [1]. However, citrus fruit is susceptible to pathogens during postharvest storage, transportation, and marketing [2,3]. The main pathogens that infect citrus fruit include *Penicillium digitatum, Penicillium italicum*, and *Geotrichum citri-aurantii*, which cause green mold, blue mold, and sour rot, respectively [3,4,5]. The pathogens have seriously affected the yield and quality of citrus fruit, and the annual losses of citrus fruit due to pathogen infection are very large [6,7,8].

*G. citri-aurantii* is the pathogen of citrus sour rot, a disease that is more difficult to control than green mold and blue mold [6,9,10]. Conventional synthetic chemical fungicides are mainly used to control postharvest diseases in citrus [11]. Commercial fungicides, such as imazalil, thiabendazole, pyrimethanil, and fludioxonil, have a good effect on inhibiting *P. digitatum* and *P. italicum*, while they have little effect on inhibiting *G. citri-aurantii* [12,13]. Sodium o-phenylphenate (SOPP), propiconazole, and guazatine can effectively inhibit *G. citri-aurantii*, but problems such as environmental pollution and fungal resistance have arisen after long-term use [14,15]. Therefore, it is very urgent to find effective alternative methods to control sour rot in citrus.

DMDC is a food additive that can be added to a variety of foods for preservative effects. DMDC has been widely used in the beverage and wine industry, and it can effectively inhibit the proliferation of microorganisms when added to beverages and wine [16]. In addition, DMDC was also applied in the fermentation of litchi juice, and the results showed that its application could ensure the microbial safety, sensory and nutritional consistency of fermented litchi juice before fermentation [17]. The use of DMDC in the field of vegetable and fruit preservation has also attracted attention. Cheng et al. [18] found that DMDC could effectively inhibit the microorganisms on the surface of Chinese cabbage. Although DMDC treatment affected the quality of appearance and texture early in storage, no significant impact was made on the nutritional composition [19].

However, whether DMDC can effectively inhibit *G. citri-aurantii* and its potential antifungal mechanism remain unclear. Therefore, the objective of this study was to determine the inhibitory effects of DMDC against *G. citri-aurantii* in vitro and in vivo. In addition, the spore germination, cell membrane integrity, ROS accumulation, MDA content, mycelial morphology, and cell ultrastructure of *G. citri-aurantii* treated with DMDC were studied. The POD and PAL activities of citrus treated with DMDC were also determined. Through all the above experiments, we look forward to understanding its potential antifungal mechanism against *G. citri-aurantii*.

## 2. Materials and Methods

### 2.1. Pathogen

The strain of *G**. citri-aurantii* used in this study was isolated from Newhall navel orange (*Citrus synesis* (L.)) in Anyuan city, Jiangxi Province, China. It was identified by morphological and molecular methods, and the pathogenicity was determined by inoculation of the fruit [20]. The fungus was cultured on potato dextrose agar (PDA:1 L of an infusion from potatoes containing 20 g/L glucose and 15 g/L agar) medium in a Petri dish (90 mm) at 28 °C for 5–7 days or potato dextrose broth (PDB: 1 L of an infusion from potatoes containing 20 g/L glucose) in a beaker flask with 180 rpm at 28 °C for 1–3 days.

### 2.2. Food Additive and Solution Preparation

Dimethyl dicarbonate (Velcorin^®^, CAS-NO. 4525-33-1, Figure 1) was purchased analytically pure from Shanghai Yuanye Biotechnology Co., Ltd., Shanghai, China with a purity of 98% and stored at 4 °C. It was dissolved in absolute ethanol to obtain DMDC solution (250 mg/mL) and diluted with sterile water to different concentrations. The solution was used immediately after preparation.

### 2.3. Fruit

Satsuma mandarin (*Citrus reticulata Blanco cv. Unshiu*) was harvested in the mature period from the citrus orchard of Huazhong Agricultural University. The citrus fruits were stored at 4 °C for approximately one week for subsequent experiments.

### 2.4. Effect of DMDC on Fungal Growth In Vitro

The fungicidal activity of DMDC was tested by dose–response curves using sterile 96-well plates as previously described [21] with minor modifications. The conidial suspension (1.0 × 10^6^ spores mL^−1^) was prepared with PDB medium, and DMDC was then added to obtain concentrations of 0, 31.25, 62.5, 125, 250, 375, and 500 mg/L. To eliminate the interference of absolute ethanol (EA), the same amount of absolute ethanol was added into the medium as a control. Then, 200 μL mixtures were transferred to the well of a sterile 96-well plate and incubated at 28 °C for 4 days. Fungal growth was determined by measuring OD_600_ values with a Multiskan Spectrum microplate spectrophotometer (BioTek Instruments, Inc., Winooski, VT, USA). The minimum inhibitory concentration (MIC) was defined as the lowest DMDC concentration that could completely inhibit fungal growth after incubation for 4 days. Then, 50 μL mixtures from each treatment were spread onto PDA plates and incubated at 28 °C. The minimum fungicidal concentration (MFC) was defined as the lowest DMDC concentration at which no colony growth occurred on the PDA plate. The experiment was repeated three times.

### 2.5. Effect on the Spore Germination of G. citri-aurantii

The conidial suspension (1.0 × 10^6^ spore mL^−1^) was added to PDB medium containing DMDC at concentrations of 0, 125, 250, and 500 mg/L. The mixtures were incubated at 28 °C for 8 h with shaking at 180 rpm. Subsequently, the germination of spores was observed and counted using an optical microscope. Spore germination was defined as the length of the germ tube exceeding the spore diameter [22]. The experiment was repeated three times.
$$\mathrm{Germination}\;\mathrm{rate}\;\left(\%\right)=\frac{\mathrm{the}\;\mathrm{number}\;\mathrm{of}\;\mathrm{germination}\;\mathrm{spores}}{\mathrm{total}\;\mathrm{number}\;\mathrm{of}\;\mathrm{spores}}\times 100\%$$

### 2.6. Effect of DMDC on Cell Membrane Permeability

Cell membrane permeability was determined by measuring the extracellular conductivity with a conductivity meter (INESA, Shanghai, China). The conidial suspension (1.0 × 10^6^ spore mL^−1^) was added to PDB medium and incubated in a shaking incubator at 28 °C and 180 rpm. After 48 h of incubation, the mycelia were collected, washed twice, and resuspended in 20 mL of sterilized distilled water. Then, DMDC was added to obtain final concentrations of 0 and 250 mg/L. The conductivity was measured by a conductivity meter at 0, 4, 8, 12, and 24 h. Finally, the conductivity was also measured after boiling for 5 min and cooling to room temperature. The relative conductivity was calculated using the following formula:$$\mathrm{Relative}\;\mathrm{conductivity}\;\left(\%\right)=\frac{\mathrm{Lx}-L0}{\mathrm{Ld}-L0}\times 100\%$$

(L0, Lx, and Ld represent the conductivity at 0 h, at 0, 4, 8, 12, and 24 h, and after boiling, respectively.)

### 2.7. Effect of DMDC on Cell Membrane Integrity

The mycelia were collected and resuspended in sterilized distilled water. DMDC was added to obtain final concentrations of 0 and 250 mg/L. The mixtures were incubated at 180 rpm and 28 °C for 8 h. Then, PI (Coolaber Technology Co., Ltd., Beijing, China) was used to detect the cell membrane integrity. After staining at 37 °C for 30 min, the mycelium was collected, washed two times, and resuspended in PBS (pH 7.2, Coolaber Technology Co., Ltd., Beijing, China), followed by detection of the fluorescence signals using a fluorescence microscope.

### 2.8. Detection of Intracellular ROS Levels and MDA Content

The culture conditions and treatment of *G. citri-aurantii* were the same as described in Section 2.7. DCHF-DA was used to detect the intracellular ROS levels. Then, the fluorescence signal of the mycelium was detected using a fluorescence microscope to reflect the intracellular ROS levels.

The MDA content of *G. citri-aurantii* treated with DMDC was determined as previously described [23] with minor modifications. The mycelia were collected and treated with 250 mg/L DMDC, followed by incubation at 180 rpm and 28 °C. Finally, the samples were taken (at 0, 4, 8, 12, and 24 h) to determine MDA content using the MDA content assay kit (Beijing Boxbio Science & Technology Co., Ltd., Beijing, China).

### 2.9. Scanning Electron Microscopy (SEM)

*G. citri-aurantii* was inoculated in PDB medium and incubated at 180 rpm and 28 °C for 48 h. Then, DMDC solution was added to final concentrations of 0 and 250 mg/L. After incubation for another 24 h, the sample was collected and fixed with 2.5% (*v*/*v*) glutaraldehyde overnight. The following preparations were performed in the electron microscope laboratory. In detail, the samples were washed thrice with PBS, postfixed for 2 h in 1% osmic acid solution, and washed with PBS three times for 15 min. Then, they were dehydrated in an ethanol series (30%, 50%, 70%, 95%, and 100%, *v*/*v*) for 20 min, two times for each concentration. After drying, the samples were coated with gold films [24]. The mycelial morphology of *G. citri-aurantii* was observed using a JEOL JSM-6390LV SEM (JEOL, Tokyo, Japan).

### 2.10. Transmission Electron Microscopy (TEM)

The treatment of *G. citri-aurantii* and collection of mycelia were the same as described in Section 2.9. The samples were fixed with 2.5% (*v*/*v*) glutaraldehyde overnight and postfixed in 1% osmic acid solution for 2 h. Then, they were dehydrated by a graded ethanol series (30–100%, *v*/*v*) and embedded in Spurr’s low-viscosity resin for 48 h at 45 °C. Thin sections (60 nm thickness) were made with a Leica Ultracut RM2265 (Leica, Vienna, Austria) and stained with lead citrate (10 min) and 2% uranyl acetate (30 min). Finally, the cell ultrastructure of the samples was observed using a JEOL H-7650 TEM (Hitachi High-Technologies Co., Tokyo, Japan).

### 2.11. In Vivo Experiment

Satsuma mandarin fruits of the same size and with no mechanical damage were selected for in vivo experiments. The fruits were soaked in a 2% sodium hypochlorite solution for 2 min, rinsed with distilled water, and dried in air [25]. Two wounds (5 mm in size and 2 mm in depth) were punctured with a wound-making tool along the equator of the fruit [26]. The DMDC solution was mixed with the conidial suspension (1.0 × 10^7^ spore mL^−1^), resulting in DMDC concentrations of 500 and 1250 mg/L, respectively (absolute ethanol contains no more than 0.4%). As a control, an equal amount of ethanol aqueous solution was mixed with the conidial suspension. Immediately, 10 μL of the mixtures was inoculated into the wounds of fruits. Each treatment had 12 fruits. All fruits were kept in plastic boxes and incubated at room temperature and 90% relative humidity for 7 days. Finally, the incidence of sour rot disease and the disease diameter were calculated for each treatment according to the method reported by [27]. The experiment was repeated three times.

We also carried out a storage experiment in Guilin, Guangxi Province. DMDC solution (99%, food grade) with a concentration of 1000 mg/L was prepared, and then citrus (*W. Murcott*) were soaked in it for 30 s. They were soaked in water as the control. These citrus were dried outdoors and stored in a simple storage room for 120 days. The weight loss rate, coloration index, and total soluble solids content of citrus were determined every 20 days.

### 2.12. Determination of Enzyme Activity in Citrus

Two wounds were made for each fruit, and 10 μL DMDC at a concentration of 500 mg/L was added to each wound. An equal amount of absolute ethanol was dissolved in water as a control. The fruits were placed in plastic boxes with 95% humidity and room temperature. Samples (0, 1, 2, 3, 4, 5, and 6 days) were taken at the edge of the wound (approximately 1 cm width) and ground with liquid nitrogen rapidly. The activities of POD and PAL were determined using an enzyme activity assay kit (Nanjing Jiancheng Bioengineering Institute, Nanjing, China).

### 2.13. Statistical Analysis

All data from the study were analyzed using SPSS 28.0 software (SPSS Inc., Chicago, IL, USA) [28]. Duncan’s multiple range tests and Student’s t-test were used to determine the significant differences at *p* < 0.05.

## 3. Results

### 3.1. DMDC Inhibited G. citri-aurantii Mycelial Growth In Vitro

The fungicidal activity of DMDC against *G. citri-aurantii* was determined in vitro. The results shows that DMDC dramatically inhibited the growth of *G. citri-aurantii*. As shown in Figure 2a, after incubation for 4 days, fungal growth was completely inhibited at a concentration of 250 mg/L (the values in Figure 2a shown in 250, 375, and 500 are initial values), indicating that the MIC value of DMDC was 250 mg/L. No fungi were observed on the PDA plate cultured at 28 °C for another 4 days when the DMDC concentration was 250 mg/L (Figure 2b). Therefore, the MFC value of DMDC was also 250 mg/L.

### 3.2. Effect of DMDC on Spore Germination

Spore germination of *G. citri-aurantii* treated with DMDC was observed in this experiment. Through preliminary experiments, we determined that it was most appropriate to observe spore germination at 8 h of incubation (not described in this paper). After incubation for 8 h, the spore germination rate of the control group reached 92.66 ± 0.58%. DMDC (125 mg/L) significantly inhibited the spore germination of *G. citri-aurantii*, with a conidial germination rate of 5.87 ± 0.37%. The inhibition of spore germination was more obvious at DMDC concentrations of 250 and 500 mg/L, with germination rates of 3.67 ± 0.11% and 2.41 ± 0.36%, respectively (Figure 3).

### 3.3. Effect of DMDC on the Cell Membrane Permeability of G. citri-aurantii

The cell membrane permeability of *G. citri-aurantii* was determined by measuring the relative conductivity. As shown in Figure 4, the relative conductivity of *G. citri-aurantii* treated with 250 mg/L DMDC continually increased throughout the period, whereas the electrical conductivity in the control group changed by a small amount. The results indicated that DMDC significantly increased the cell membrane permeability.

### 3.4. Effects of DMDC on the Cell Membrane Integrity of G. citri-aurantii

When the integrity of the cell membrane is damaged, propidium iodide (PI) enters the cell. Therefore, cell membrane integrity could be determined by detecting fluorescence signals after PI staining. As shown in Figure 5a,b, PI staining results showed no fluorescence in the control group. However, fluorescence signals were detected in 35.78% of mycelia after treatment with 250 mg/L DMDC, indicating that DMDC damaged cell membrane integrity.

### 3.5. Effects of DMDC on Intracellular ROS Levels and MDA Content from G. citri-aurantii

The intracellular ROS levels could be determined by DCHF-DA staining, and they were positively correlated with fluorescence intensity. As shown in Figure 5c,d, almost no fluorescence signal of DCHF-DA was detected in the control group, but a large number of fluorescence signals were detected in the DMDC treatment group. The results suggested that DMDC at 250 mg/L led to the accumulation of intracellular ROS.

The lipid peroxidation of mycelium was determined by the measurement of MDA content. We found that the MDA content in the control group was maintained at a low level, but the MDA content treated with 250 mg/L DMDC was higher than that in the control group from 4 h onwards (Figure 6), indicating that DMDC treatment led to lipid peroxidation.

### 3.6. Effect of DMDC on Mycelium Morphology

The morphology of mycelium could be observed by SEM. We found that the mycelium of the control group was smooth and full (Figure 7a), but the hyphae were folded and collapsed after treatment with 250 mg/L DMDC (Figure 7b), indicating that DMDC changed the original morphology of mycelia.

### 3.7. Effect of DMDC on the Cell Ultrastructure

The cell ultrastructure was observed by TEM. The results are shown in Figure 7c–f. As shown by the red arrow, the internal structure of mycelia in the control was clear and complete (Figure 7c,e). However, after DMDC treatment, with the damage to mycelium organelles, vacuoles were significantly increased, and cell membrane edges were blurred (Figure 7d,f).

### 3.8. In Vivo Experiments

As shown in Figure 8, the disease incidence and the disease diameter were reduced after DMDC transient treatment compared with the control after inoculation for 7 days. In detail, the disease incidences were 15.28% and 5.56% when the fruits were treated with 500 and 1250 mg/L DMDC, respectively, whereas the control was 100%. In addition, the disease diameter was 51.98 ± 6.09 mm with the control, but the disease diameters were 23.27 ± 9.27 and 9.25 ± 1.85 mm when treated with 500 and 1250 mg/L DMDC, respectively.

In addition, the storage experiment results are shown in Table 1. DMDC did not change the fruit weight loss rate, coloration index, or total soluble solids compared with the control group during storage.

### 3.9. Determination of Citrus Enzyme Activity

The activities of POD and PAL were determined in the study. As shown in Figure 9a, the POD activity of citrus treated with DMDC was significantly increased compared with that of the control and peaked on the first day. In addition, DMDC also significantly increased the PAL activity in fruits, which remained at a higher level after the second day compared with the control (except on the fifth day).

## 4. Discussion

In the present study, we determined the inhibitory effect of DMDC on *G. citri-aurantii* in vitro and in vivo. The results showed that the growth of *G. citri-aurantii* was completely inhibited when the DMDC concentration was 250 mg/L (Figure 2). DMDC at a concentration of 250 mg/L significantly inhibited the spore germination of *G. citri-aurantii*, and the germination rate was 3.67 ± 0.11%, while that of the control group was 92.66 ± 0.52% (Figure 3).

Many studies have reported that the destruction of cell membranes and cell walls are the antifungal mechanisms of many antifungal substances and antagonists [23,27]. OuYang et al. [29] found that the combined use of citronellal and cinnamaldehyde accelerated damage to the cell wall and cell membrane; thus, inhibiting the growth of *Penicillium digitatum*. He et al. [30] found that natamycin could damage the plasma membrane of *Botrytis cinerea* and *Penicillium expansum*, leading to the release of intracellular contents and eventual cell death. Our results showed that 250 mg/L DMDC continuously increased the extracellular relative conductivity of *G. citri-aurantii* from 0 to 24 h, while the change in the control group was small (Figure 4). This result indicated that DMDC increased the permeability of the *G. citri-aurantii* cell membrane. This continued increase in membrane permeability might have more serious effects on the cell, such as leakage of intracellular macromolecules [25]. Furthermore, the PI staining experiment was also carried out to determine the membrane integrity of *G. citri-aurantii*. PI is a nuclear staining reagent that can stain DNA. It cannot pass through intact cell membranes but can pass through broken cell membranes to stain nuclei. As shown in Figure 5a,b, a large number of red fluorescence signals were observed after *G. citri-aurantii* was treated with 250 mg/L DMDC, indicating that DMDC damaged the cell membrane integrity of *G. citri-aurantii*.

The destruction of cell membranes by some antifungal substances may be related to lipid peroxidation [23,31]. MDA content evaluates the level of lipid peroxidation and indirectly reflects the severity of cell damage [27,32]. Therefore, we determined the MDA content of *G. citri-aurantii* treated with 250 mg/L DMDC using an MDA content assay kit. Our results suggested that the MDA content of *G. citri-aurantii* treated with DMDC was significantly higher than that of the control group (Figure 6), which indicated that the damage to the cell membrane might be caused by lipid peroxidation. The accumulation of ROS can cause oxidative damage to cell components and result in impaired cell function and other adverse effects [33]. Polyunsaturated fatty acids are the preferred targets for ROS due to the presence of numerous bis-allylic hydrogen atoms that are easily oxidized and form lipid peroxidation products, such as malondialdehyde [34]. We determined the level of intracellular ROS by DCHF-DA staining. As revealed in Figure 5c,d, 250 mg/L DMDC caused the accumulation of intracellular ROS in *G. citri-aurantii*. Therefore, we conjectured that the accumulation of intracellular ROS led to lipid peroxidation and damaged cell membranes.

The mycelium morphology could be observed by SEM. We found that the mycelium was folded and collapsed after treatment with 250 mg/L DMDC compared with the control (Figure 7a,b). This was consistent with the results of [9,23]. The cell ultrastructure could be observed by TEM. Cheng found that cinnamic acid (CA) treatment of *G. citri-aurantii* caused damage to organelles and empty cavities were observed in the cells [9]. Polyhexamethylene biguanide (PHMB) and polyhexamethylene guanide (PHMG) treatment of *G. citri-aurantii* also had the same results [12]. In the experiment, we found that DMDC caused damage to the subcellular structure of *G. citri-aurantii*, such as mitochondria, and vacuolation could be observed. The cell membrane profile of the treated group was less clear than that of the control group (Figure 7c,f). This result further proved that DMDC could inhibit *G. citri-aurantii*.

In vivo experiments were also conducted in the study [11], and citrus fruits were inoculated with *G. citri-aurantii* and DMDC. After 7 days of inoculation, 500 mg/L DMDC reduced fruit incidence and disease diameter by 84.72% and 55.23%, respectively, and 1250 mg/L DMDC reduced fruit incidence and disease diameter by 94.44% and 82.2%, respectively, compared with the control group (Figure 8). This finding indicates that DMDC transient treatment can effectively inhibit the pathogenicity of *G. citri-aurantii.* Previous studies have found that DMDC can also inhibit the incidence of blue mold, when the concentration is 7.5 mM (1.005 g/L), the inhibition rate is approximately 20% [35]. Wu et al. [36] found that the application of wax with cinnamaldehyde (WCA) significantly reduced the incidence of sour rot on citrus fruit. The incidence of the fruit treated with 2 mL L^−1^ WCA was only 50%, while the control fruit was 100% after 8 days of storage. Wang et al. [11] found that the treatment method in which conidia were mixed with the antimicrobial peptide PAF56 and incubated for 16 h showed the highest degree of control of *G. citri-aurantii* growth on oranges. It could significantly reduce the incidence of sour rot by 34% or more. Compared with the above methods, DMDC could control sour rot more effectively. In addition, we carried out a storage experiment in Guilin, Guangxi Province. The results showed that DMDC treatment did not affect fruit quality, including weight loss rate, coloration index, and total soluble solids.

Improving fruit resistance is also a strategy to inhibit the growth of pathogens in vivo. POD is considered to be associated with disease resistance in different plant–pathogen interactions [33]. PAL exists widely in plants and plays an important role in normal plant growth and resistance to pathogen invasion [37,38]. We measured the activities of POD and PAL in fruits treated with DMDC. The data showed that DMDC treatment increased POD and PAL activity in fruits compared with the control (Figure 9). The results indicated that DMDC could inhibit the growth of *G. citri-aurantii* in citrus by increasing the activities of POD and PAL.

## 5. Conclusions

We confirmed that *G. citri-aurantii* could be effectively inhibited by DMDC in vitro. The antifungal mechanism might be that DMDC destroyed the organelles (such as mitochondria) and caused ROS accumulation and lipid peroxidation; thus, damaging cell membranes and finally inhibiting *G. citri-aurantii*. Meanwhile, DMDC transient treatment of *G. citri-aurantii* could effectively inhibit its pathogenicity, and the POD and PAL activities of citrus fruits could be promoted by DMDC. In the future, molecular biology and other methods will be used to further clarify the action mechanism of DMDC against *G. citri-aurantii* and find the target of DMDC acting on *G. citri-aurantii* to provide a basis for the development of new fungicides.

## Figures and Tables

**Figure 1 foods-11-02328-f001:**
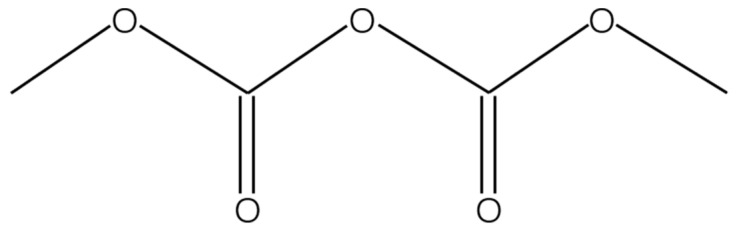
The structure of dimethyl dicarbonate.

**Figure 2 foods-11-02328-f002:**
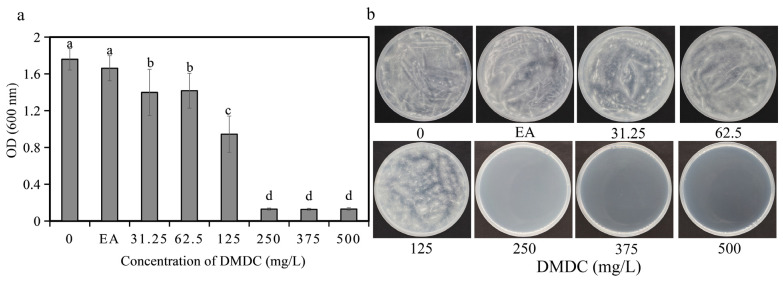
Effect of DMDC on fungal growth in vitro. (**a**) Dose–response curves of growth inhibition of *G. citri-aurantii* by DMDC in vitro. There were three replicates of the samples, and the data were the mean ± standard deviation of OD_600_ measurements of *G. citri-aurantii* treated with DMDC after incubation for 4 days. Different letters on each column represent significant differences (*p* < 0.05) according to Duncan’s test. (**b**) The growth of *G. citri-aurantii* on PDA plates after treatment with DMDC for 4 days.

**Figure 3 foods-11-02328-f003:**
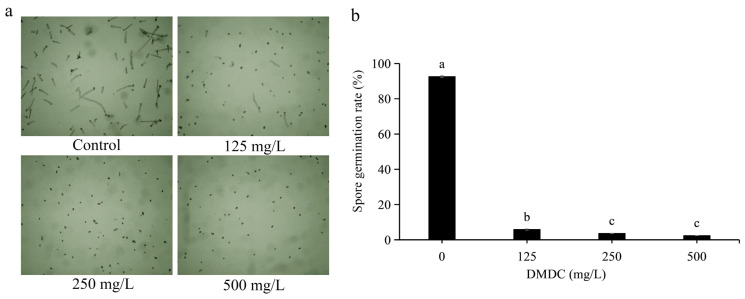
*G. citri-aurantii* was treated with DMDC at concentrations of 0, 125, 250, and 500 mg/L, and germination was observed after 8 h. (**a**) Spore germination was observed under a light microscope. (**b**) Spore germination rate after treatment with different concentrations of DMDC. The data represent the mean ± standard deviation (SD). Different letters on each column represent significant differences (*p* < 0.05) according to Duncan’s test.

**Figure 4 foods-11-02328-f004:**
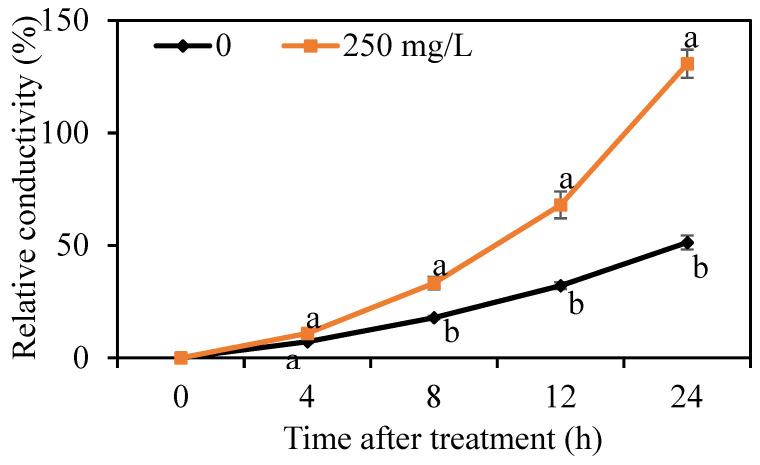
Effect of DMDC on the cell membrane permeability of *G. citri-aurantii*. The relative conductivity of *G. citri-aurantii* treated with 250 mg/L DMDC and untreated at 0, 4, 8, 12, and 24 h. The data represent the mean ± standard deviation (SD). Different letters on each column represent significant differences (*p* < 0.05) according to a t-test.

**Figure 5 foods-11-02328-f005:**
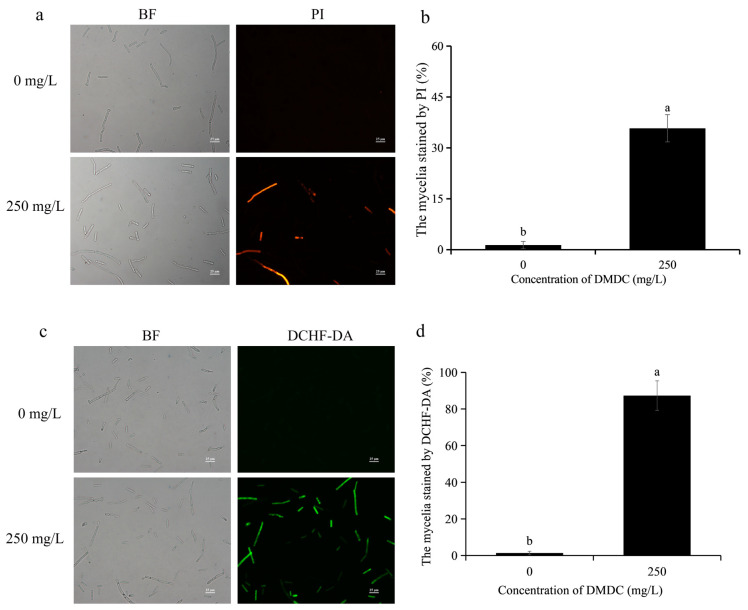
Effects of DMDC treatment on cell membrane integrity and ROS accumulation of *G. citri-aurantii*. *G. citri-aurantii* stained by PI (**a**) and DCHF-DA (**c**). Bars = 25 μm; fluorescence signals were detected in 35.78% of mycelia after treatment with 250 mg/L DMDC (**b**). A large number of fluorescence signals were detected after DMDC treatment (**d**). Different letters on each column of (**b**,**d**) represent significant differences (*p* < 0.05) according to a t-test.

**Figure 6 foods-11-02328-f006:**
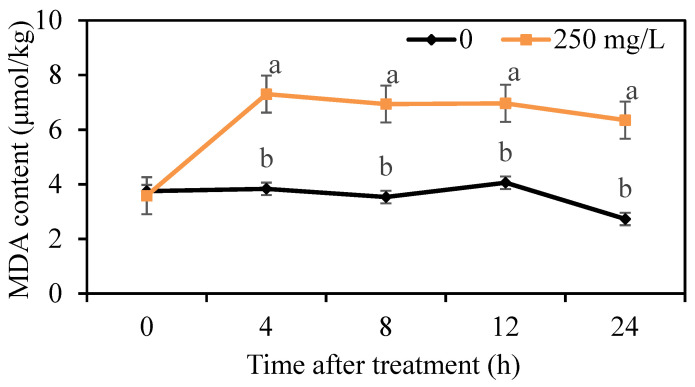
Effects of DMDC treatment on the MDA contents of *G. citri-aurantii*. The MDA content of *G. citri-aurantii* treated with 250 mg/L DMDC and untreated was determined at 0, 4, 8, 12, and 24 h. The data represent the mean ± standard deviation (SD). Different letters on each column represent significant differences (*p* < 0.05) according to a t-test.

**Figure 7 foods-11-02328-f007:**
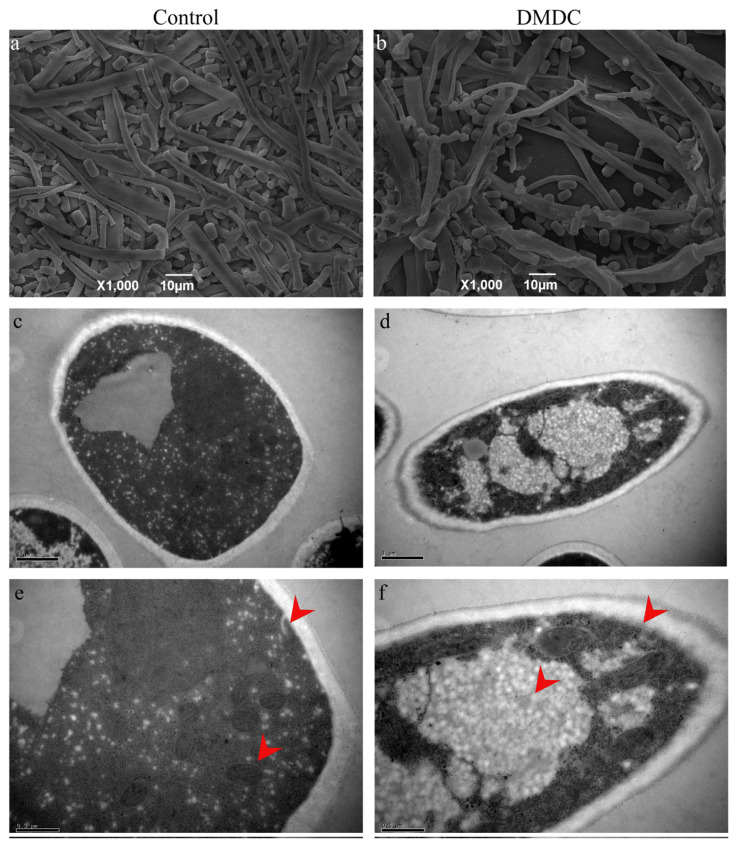
SEM and TEM observation of *G. citri-aurantii* treated with 250 mg/L DMDC. SEM observation in the control group (**a**) and treated with 250 mg/L DMDC (**b**); TEM observation in the control group (**c**,**e**) and treated with 250 mg/L DMDC (**d**,**f**).

**Figure 8 foods-11-02328-f008:**
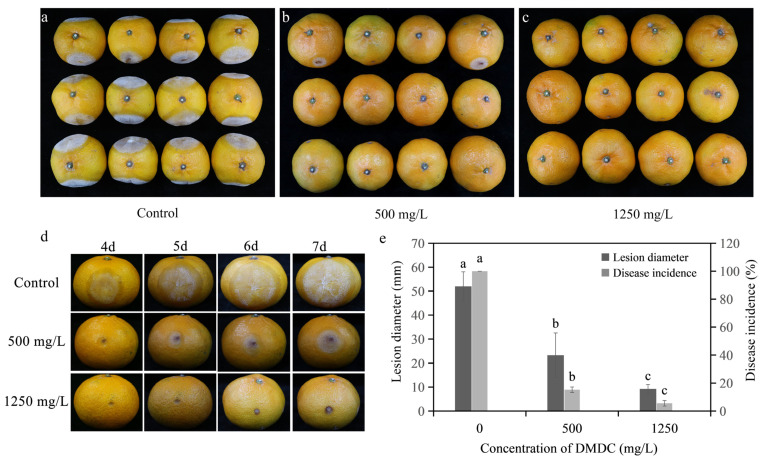
Inhibitory effect of DMDC against *G. citri-aurantii* on citrus fruits. (**a**–**c**) The incidence of all fruits with different treatments at 7 days after inoculation; (**d**) the fruits of different treatments after 4–7 days of inoculation; (**e**) the effects of DMDC treatment on fruit disease incidence and disease diameter. The data represent the mean ± standard deviation (SD). Different letters on each column represent significant differences (*p* < 0.05) according to Duncan’s test.

**Figure 9 foods-11-02328-f009:**
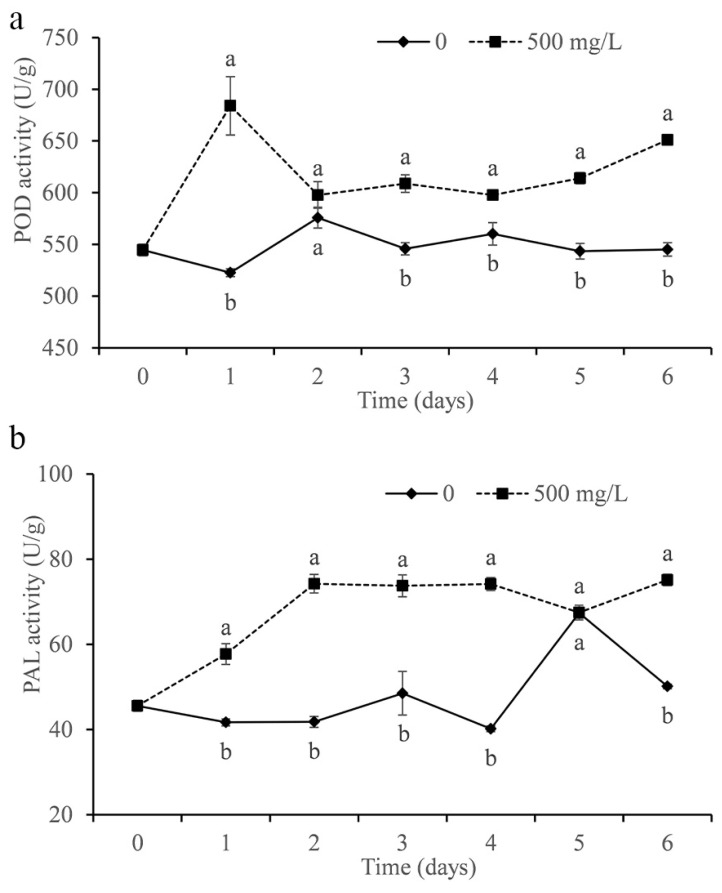
Effect of DMDC (500 mg/L) treatment on the activities of POD (**a**) and PAL (**b**) in citrus fruit. The data represent the mean ± standard deviation (SD). Different letters on each column represent significant differences (*p* < 0.05) according to a t-test.

**Table 1 foods-11-02328-t001:** Results of fruit quality determination after DMDC treatment.

The Physiological Indicators	Treatments	Days after Storage					
		20	40	60	80	100	120
Weight loss rate (%)	Control	0.79 ± 0.07 Ae	1.63 ± 0.14 Ad	2.90 ± 0.23 Ac	3.14 ± 0.26 Ac	3.88 ± 0.34 Ab	4.46 ± 0.38 Aa
	DMDC	0.83 ± 0.09 Ae	1.69 ± 0.15 Ad	2.98 ± 0.25 Ac	3.26 ± 0.28 Ac	4.05 ± 0.36 Ab	4.70 ± 0.43 Aa
Coloration index	Control	11.47 ± 1.08 Aab	12.08 ± 1.63 Aa	11.42 ± 0.92 Aab	11.14 ± 1.02 Ab	11.21 ± 1.13 Aab	11.02 ± 1.12 Ab
	DMDC	11.80 ± 1.05 Abc	12.77 ± 1.23 Aa	12.16 ± 0.90 Aab	11.87 ± 1.18 Aab	10.85 ± 0.86 Ac	11.03 ± 1.40 Abc
Total soluble solids (%)	Control	10.34 ± 1.23 Aa	11.42 ± 0.56 Aa	11.60 ± 0.76 Aa	10.88 ± 0.91 Aa	11.26 ± 0.80 Aa	10.86 ± 0.66 Aa
	DMDC	10.88 ± 0.72 Aa	11.72 ± 0.63 Aa	11.60 ± 0.64 Aa	11.56 ± 0.61 Aa	11.68 ± 1.55 Aa	10.82 ± 0.32 Aa

The data represent the mean ± standard deviation (SD). Different capital letters represent significant differences (*p* < 0.05) between different treatments. Different lowercase letters represent significant differences (*p* < 0.05) between different times.

## Data Availability

The data presented in this study are available on request from the corresponding author. The data are not publicly available at this time as the data also forms part of an ongoing study.

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
