# Peer review of "Dimethyl Dicarbonate as a Food Additive Effectively Inhibits Geotrichum citri-aurantii of Citrus"

_foods, 2022, doi:10.3390/foods11152328_

Round 1

Reviewer 1 Report

Abstract – Every acronym must be introduced in its first appearance (e.g., DCHF-DA). Those acronyms not used in the abstract may be introduced in the subsequent section.

Keywords – Replace those already in the title.

L30-32 – There are recent statistics on citrus production, which must be updated in the introduction. The authors must also mention the leading producers.

L52 – Replace “the application of DMDC” with “its application.”

L54 – “The use of DMDC…”

L55 – Please indicate each reference number right after the authors, as indicated in the Guide for authors.

L74 – Degree Celsius must be reviewed throughout the manuscript.

L106 – The authors must standardize units according to the SI (e.g., “h” instead of “hours,” “min’ instead of “minutes”)

L133 – Please do not start sentences with Roman figures.

L199-206 – Graphs in Figures 2, 3, 4, 5, 6, and 8 have too low resolution. I suggest the authors copy and paste the metafile from SPSS into the Word document, so the graph quality is maintained. In addition, plate and SEM images must be enhanced to >300 dpi.

L280 – Unnecessary explanation; this must be indicated in the M&M section.

L322-324 – Same comment as L55.

L389 – Please remove “Through the series of studies above” and go straight to the conclusions. Also, use passive voice, as it sounds more professional.

Reviewer 2 Report

Dear authors,

you submitted a manuscript on the fungicidal effects of DMDC on Geotrichum citri-aurantii (sour rot of citrus).  The main goal is to find aletranives for synthetic chemical fungicides to control this postharvest disease. The study is divided in to four sections: (1) In vitro efficacy experiments, (2) morphological responses of the fungus, (3) an in vivo experiment, and (4) physiological responses of fruits after treatment.

It is known that DMDC has antimicrobial effects and that it could control green and blue mold on citrus. New is that it is efective in in vitro experiments. New is the morphological changes in the fungus. New is also the physiological response of citrus (POD, PAL). To my knowledge there is only one study published on the effect of DMDC on POD, but in chinese cabbage. You could also show that higher concentrations of DMDC (1000 mg / L) did not affect the storage of fruits.

But the presented in vivo experiment on fruits seems not suited to address the question, if DMDC is an effective postharvest controlmeasure for sour rot. I have made a detaied comment in the attached PDF on this experiment. My concern is that with your approach to mix DMDC with the inoculum suspension before you apply it to wounds is not able to show a protective or curative efficacy. The reason: it is more than unlikely under practical conditions that DMDC and the fungus will "arrive" at the fruit at the very same time. Here protective or a curative treatment should have been made.

I was also wondering why you have used higher concentrations in some experiments, including the in vivo experiment but also the physiological response of citrus. Here you use 500 or 1000 mg / l, instead of 250 mg / L as you reported to be the effectve minimal concentration. It woukd be more coherent to use the same concentration in all experiments for better comparison of the effects. Now it remains unclear, if an effect could have been reached with 250 mg.

You should clarify and explain this more.

See also my specific comments made in the attched PDF.

Round 2

Reviewer 1 Report

Figures still lack quality. The only figures that were really improved were Figures 4 and 6. When the reader goes to zoom in on the figures, it is clear that they are low resolution, and some details are missed. In addition, the text needs to be proofread, as some typos were detected.
